# Seroprevalence of COVID-19 infection among vaccine naïve population after the second surge (June 2020) in a rural district of South India: A community-based cross-sectional study

Carolin Elizabeth George[1], Leeberk Raja Inbaraj[1]*, Shon Rajukutty[1], Roshni Florina Joan[1], Arun Karthikeyan Suseeladevi[2], Sangeetha Muthuraj[1], Sindhulina Chandrasingh[3]

1 Division of Community Health and Family Medicine, Bangalore Baptist Hospital, Bangalore, Karnataka, India, 2 Division of Gastrointestinal Sciences, The Wellcome Trust Research Laboratory, Christian Medical College, Vellore, Tamil Nadu, India, 3 Department of Microbiology, Bangalore Baptist Hospital, Bangalore, Karnataka, India

* leeberk2003@gmail.com

## Abstract

### Objective

To determine the seroprevalence of the SARS Cov 2 infection among vaccine naive population in a rural district of South India post-second surge.

### Methodology

We conducted a cross-sectional study in the five villages of a randomly chosen sub-district in the Bangalore rural district. We did house to house surveys and recruited 831 vaccine naive adults in July 2021. We tested samples for the presence of antibodies (including IgG & IgM) to SARS CoV-2 using the Roche Elecsys SARS-CoV-2 –S assay that quantifies antibodies against the receptor-binding domain (RBD) of the spike (S) protein.

### Results

We estimated an overall prevalence of 62.7% (95% CI: 59.3–66.0) and an age-and gender-adjusted seroprevalence of 44.9% (95% CI: 42.5–47.4). When adjusted for test performance, the seroprevalence was 74.64% (95% CI: 70.66–78.47). The case-to-undetected-infected ratio (CIR) was 1: 8.65 (95% CI 1:8.1–1:9.1), and the Infection Fatality Rate (IFR) was 16.27 per 100,00 infections as of 13 July 2021. A history of at least one symptom suggestive of COVID-19 or a positive COVID-19 test of self or a family member in the past were significantly associated with seropositivity.

**Data Availability Statement:** All relevant data are within the manuscript and its Supporting Information files.

**Funding:** The authors received no specific funding for this work.

**Competing interests:** The authors have declared that no competing interests exist.

## Conclusion

We report a high seroprevalence of COVID-19 infection despite the advantages of low population density and well-ventilated landscapes in rural areas. CIR and IFR were higher than the previous serosurvey conducted in the same population during the first surge. The thought of achieving herd immunity comes with relief. However, it's vital to put efforts into building population health and rural health infrastructure to avert future health catastrophes.

## Introduction

The second wave of the Covid 19 pandemic swept across the whole of India at a raging pace, devastating lives and livelihoods. In India, the peak of the first wave was in September 2020 with a subsequent reduction in daily cases until February 2021 [1]. Pandemic fatigue, emergence of mutant strains, political and religious mass gatherings and vaccine complacency resulted in a massive second surge. The slow rising curve of new cases in February 2021 soon transformed into a steep uphill by April, 2021 [2]. On 19 April 2021, the number of new cases was about 0.3 million, which was already three times that of the first wave around the same month the previous year [1]. The explosion of serious cases created a shortage of hospital beds, medical equipments, oxygen, trained personnel and lifesaving drugs leading to a major breakdown of the country's healthcare system [3].

During the first surge in 2020, the infection mostly marked its prominence in thickly populated urban settlements [4]. In the second surge, rural areas also faced the brunt of the pandemic contributing to 53% of new cases and 52% of deaths by May 2021. Though these numbers indicate major spread of COVID 19 to rural India, the magnitude and the extent is still not fully known.

The Community Health Division (CHD) of Bangalore Baptist Hospital has been providing curative and preventive health services in the Bangalore Rural district for a while now. The serosurvey done during the first wave in September 2020 in this district revealed that the COVID 19 infection spared most of the rural population (12.4% Vs. 57.9%) as compared to slums [4,5]. The objective of this study is to determine the seroprevalence of the SARS Cov 2 infection in the Bangalore rural district post second surge. It is important to understand the burden of the susceptible in these areas after the second wave to evaluate the chance of a next wave and also to design health and social interventions which are relevant for this population. We also hope that the findings of this study will help the local health authorities in strategizing vaccinations and add valuable data to researchers across the globe.

## Materials and methods

### Setting

The study was carried out in the Bangalore Rural District of Karnataka, a state in Southern India. This district, which is situated in Bangalore, is divided into four taluks (sub-districts) and 105-gram panchayats (village administrative entities). A cluster of 10–15 villages occupy each gram panchayat [6]. According to the Indian Census of 2011, the population was 9,90,923, with a population density of 441 people per square kilometre. People in the district mainly depend on agriculture and related activities such as cattle rearing for their livelihood. Health care is provided by both the government health system and private practitioners. In addition to providing primary health care, the community outreach of Bangalore Baptist

Hospital has over time established strong community ties by forming village health committees and collaborating with the rural self-government.

## Study design and sample size

Based on WHO's recommendation as the most appropriate study design [7], we performed a cross-sectional sero-epidemiological survey in this Bangalore rural district. All consenting vaccine naïve adults (18 years or above) were considered eligible for the study. Assuming a seroprevalence of 40%, with an absolute precision of 5% and a design effect of 2, we computed a minimal sample size of 738 at 95% confidence interval. Estimating an average population of 600 adults (18 years old and more) in each village and accounting for vaccinations (50%), non availability (15%) and refusal (10%), we assumed that five villages are required to complete the sample size. The study was approved by the Institutional Review Board of Bangalore Baptist Hospital.

## Data collection

The Bangalore Rural District is divided into four sub-districts, each of which is further subdivided into gram panchayats, or village administrative entities. We randomly chose one subdistrict among the four, then randomly chose five villages from the selected sub district and approached all the eligible individuals in the chosen villages.

We conducted house to house visits from 2–22 June 2021 and included all the eligible family members who consented for the study. The Data collection team consisted of a physician, nurse, interviewer, phlebotomist and a field staff. If a family refused to participate or if a house was locked, the team moved to the next house. However, the locked house was revisited once again before completing one particular village. The team covered every household in each village before moving to the next village.

After obtaining a written consent, each participant was interviewed by a trained research coordinator who had prior experience in data collection, using a questionnaire. Questions on demographics (age, gender, education, comorbidities such as diabetes, hypertension, lung disease and cancer) and the history of COVID-19 infection (history of being detected positive for COVID-19, family members being positive, history of at least one symptom suggestive of COVID-19 in the last 12 months) were included in the questionnaire. The interviewer used Epi-info 7.0 TM mobile application-based tool to record replies offline which were later downloaded for analysis.

The phlebotomists drew 2ml of blood in a simple vacutainer via venepuncture from each participant after the interview was completed. The vacutainers were kept in cold chain and were transported to BBH laboratory in less than 5 hours. The serum was separated and tested for antibodies using the Elecsys Anti-SARS-CoV-2—S assay (Roche Diagnostics, Switzerland) which quantifies antibodies against the receptor-binding domain (RBD) of the SARS-CoV-2 spike (S) protein in humans [8].

The antigens within the reagent capture anti-SARS-CoV-2 (including IgG & IgM). This assay has a measuring range of 0.40 to 250 U/ml in undiluted samples with a concentration of <0.80 U/ml considered negative and > = 0.80 U/ml considered positive according to the manufacturer's instructions [8]. The Elecsys anti-SARS-CoV-2 S assay is reported to have its highest sensitivity of 84.0%% (95% CI:73.8–94.2) and specificity of 100% at 15 to 30 days post-PCR positivity and exhibited no cross-reactivity. The assay has a PPV of 100%, and an NPV between 98.3% and 99.8% at >14 days post-PCR positivity, depending on the seroprevalence estimate [9].

For statistical analysis, we used STATA 15.0 and the Statistical Package for the Social Sciences version 20.0. The unadjusted COVID-19 IgG seroprevalence was reported in percent with a 95% confidence interval (CI). We also reported total antibody titres among males and females stratified by age. We calculated weights for reporting age-and-gender standardised seroprevalence using Karnataka's rural area figures from the Sample Registration System (SRS) statistical report 2018 [10]. (S1 Table)

The case-to-undetected-infections ratio (CIR) was calculated as a ratio of the number of reported RT-qPCR-confirmed COVID-19 cases two weeks before the serosurvey imitation to the number of people in our study who have antibodies. This was based on an earlier study that found median seroconversion times for total antibodies, IgM, and then IgG at day-11, day-12, and day-14, respectively, in hospitalised patients, with IgG and IgM seroconversion occurring simultaneously or sequentially [11,12]. We used the reported number of fatalities after three weeks of the survey to estimate the plausible range of the infection fatality ratio (IFR), assuming a three-week lag time between infection and death. It was calculated by the number of deaths reported upon the total number of people with high-affinity antibodies per 10000 infections. To calculate all of these parameters, we used a projected population of 2021 Bangalore rural district based on 2011 census data prepared by the Directorate of Economics and Statistics, Bangalore 2013 [13]. Using chi-square tests, the relationship between seroprevalence and comorbid conditions and socio-demographic characteristics was investigated.

## Results

We recruited 831 participants from five villages of Devanahalli sub-district, Bangalore rural district. More than half of the participants were women (58.6%) and one third of the population were in the middle age group (31–40 yrs.). The mean age of the participants was 36.15 +/12.55 years. The most common occupation was farming (26.7%) and half of the women in the study were home makers (50.5%). A small proportion (15%) had no formal education and more than one third had more than 10 years of formal education. The most common comorbidities reported were hypertension (5.1%) and diabetes (6.3%). Among 835 participants, 17.1% reported to have at least one symptom suggestive of COVID-19 in the last 12 months prior to the survey, fever being most common symptom (8.8%) (Table 1)

The overall crude unadjusted seroprevalence among the vaccine naive population in July 2021 was 62.7% (95% CI: 59.3–66.0) (Table 2). The age and gender-adjusted seroprevalence of COVID-19 was 44.9% (95% CI: 42.5–47.4) (S1 Table). The unadjusted seroprevalence among participants with diabetes and hypertension was 69.2% (95% CI 54.9–81.3) and 71.4% (95% CI: 55.4–84.3) respectively. The antibody titres were highest in 41–60 year group in males and above 60 age group in females (Fig 1). Among 521 individuals who were seropositive, 126 (24.2%) of them had at least one symptom suggestive of COVID-19 in the last 12 months before the survey. One fifth of them (19.6%) were tested positive for COVID-19 in the past and 17.3% had at least one member tested positive for COVID-19 in the last one year. Almost one quarter (22.6%) of those who were tested positive were either admitted in a COVID care centre or in a hospital.

The cumulative number of SARS CoV-2 infection based on our adjusted prevalence and projected population in 2021 in Bangalore rural district was 520272 (95% CI: 492463–549241) during two weeks before the beginning of the study (02 June to 22 June 2021). The seroprevalence was 74.64% (95% CI: 70.66–78.47) and the cumulative number of infections was 8,64,417 (95% CI:818068–908449) when adjusted for sensitivity and specificity of the test kit.

Demographic factors such as age, gender, education, occupation or number of rooms in the house were not associated with seropositivity (p-value >0.05). However, a history of at least

**Table 1. Sociodemographic characters of the study population.**

| Demographics | Variables | N | Percentage |
|---|---|---|---|
| Age in years | 18–20 | 70 | 8.4 |
| | 21–30 | 236 | 28.4 |
| | 31–40 | 279 | 33.6 |
| | 41–50 | 150 | 18.1 |
| | 51–60 | 54 | 6.5 |
| | >61 | 42 | 5.0 |
| Sex | Female | 487 | 58.6 |
| | Male | 344 | 41.4 |
| Education | Illiterate | 125 | 15.0 |
| | Primary | 123 | 14.7 |
| | Middle or High School | 279 | 33.4 |
| | PUC/Diploma | 170 | 20.3 |
| | Degree | 138 | 16.5 |
| Occupation | Housewife | 247 | 29.6 |
| | Domestic Helper | 9 | 1.1 |
| | Daily wage labourer | 77 | 9.3 |
| | Farmer | 223 | 26.7 |
| | Not working | 136 | 16.3 |
| | Professional | 51 | 6.1 |
| | Other | 89 | 10.7 |
| Comorbidities | Diabetes | 53 | 6.3 |
| | Hypertension | 43 | 5.1 |

one self-reported symptoms suggestive of COVID-19 in the last 12 months before the study (88.7% Vs 57.3%), a positive COVID-19 test in the past (95.3% Vs 57.9%) and a past history of confirmed COVID 19 infection of a family member (88.2% Vs 59.1%) were significantly associated with seropositivity (Table 3).

The cumulative number of RT-PCR confirmed cases were 59,122 as on 23 June 2021 in Bangalore rural district. Hence, we estimated 8.65 (511722/59122) undetected infected individuals for every RT-PCR confirmed cased, i.e., case-to-undetected-infected ratio (CIR) of 1: 8.65 which could range from 1: 8.19 to 1:9.13. We also estimated the Infection Fatality Rate (IFR) as 16.27 per 100,00 infections as on 13 July 2021 based on the age-gender adjusted seroprevalence rate estimated from the study and cumulative deaths reported by the Govt. of Karnataka, India [14].

**Table 2. Unadjusted Seroprevalence of COVID-19 in Bangalore rural district, India.**

| | Category | Male | Prevalence (95% CI) | Female | Prevalence (95% CI) | Total | Overall prevalence (95% CI) |
|---|---|---|---|---|---|---|---|
| Age (yrs) | 18–20 | 39 | 59 (42.1–74.4) | 31 | 71 (52–75.8) | 70 | 64.3 (51.9–75.4) |
| | 21–30 | 72 | 69.4 (57.5–79.8) | 164 | 62.(54.9–70.2) | 236 | 64.8 (58.4–70.9) |
| | 31–40 | 111 | 74.8 (65.6–82.5) | 168 | 53 (45.1–60.7) | 279 | 61.6 (55.7–67.4) |
| | 41–50 | 75 | 61.3 (49.4–72.4) | 75 | 56 (44.1–67.5) | 150 | 58.7 (50.3–66.6) |
| | 51–60 | 26 | 53.8 (33.4–73.4) | 28 | 67.9 (47.6–84.1) | 54 | 61.1 (46.9–74.1) |
| | >60 | 21 | 57.1 (34.0–78.2) | 21 | 85.7 (63.7–97) | 42 | 71.4 (55.4–84.3) |
| | Total | 344 | 66. 3 (61–71.3) | 487 | 60.2 (55.7–64.5) | 831 | 62.7 (59.3–66) |

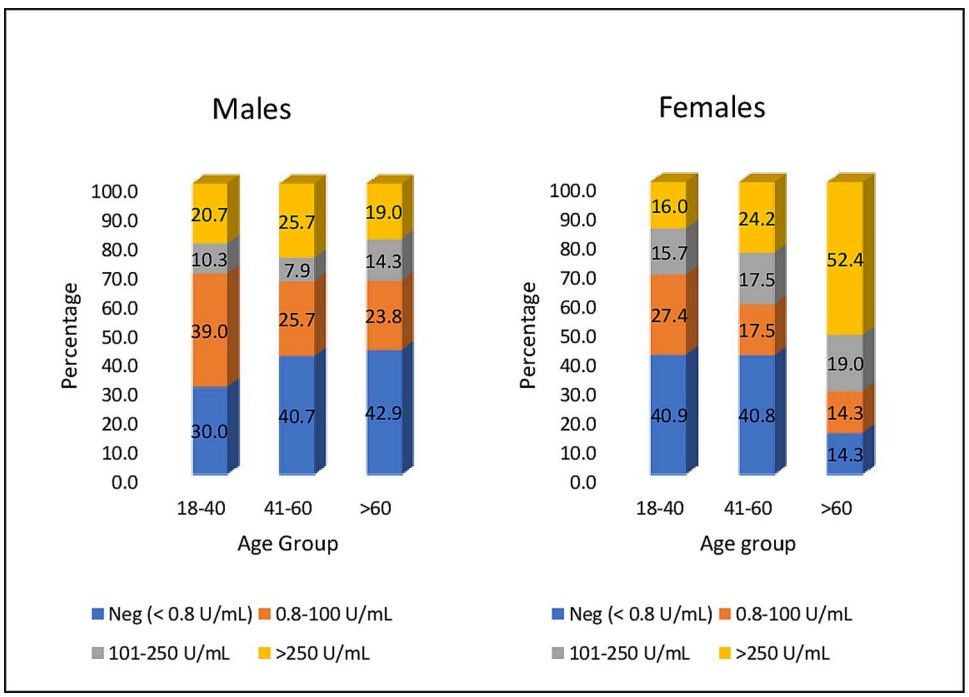

**Fig 1. Age and gender distribution of antibody titre.**

**Table 3. Factors associated with Seroprevalence.**

| Parameters | Categories | Serological status | | Total | p value |
|---|---|---|---|---|---|
| | | Non–Reactive | Reactive | | |
| Age in years | < = 60 | 298 (37.8) | 491 (62.2) | 789 | 0.23 |
| | >60 | 12 (28.6) | 30 (71.4) | 42 | |
| Gender | Male | 116 (33.7) | 228 (66.3) | 344 | 0.07 |
| | Female | 194 (39.8) | 293 (60.2) | 487 | |
| Education | Lower (< = 10 years) | 198 (37.9) | 325 (62.1) | 523 | 0.66 |
| | Higher (>10years) | 112 (36.4) | 196 (63.6) | 308 | |
| Occupation | Not working/ House wife | 137 (35.9) | 245 (64.1) | 382 | 0.42 |
| | Others | 173 (38.5) | 276 (61.5) | 449 | |
| Hypertension | Yes | 12 (28.6) | 30 (71.4) | 42 | 0.23 |
| | No | 298 (37.8) | 491 (62.2) | 789 | |
| Diabetes | Yes | 16 (30.8) | 36 (69.2) | 52 | 0.31 |
| | No | 294 (37.7) | 485 (62.3) | 779 | |
| No. of rooms in the house | < = 2 | 236 (36.8) | 405 (63.2) | 641 | 0.59 |
| | >2 | 74 (38.9) | 116 (61.1) | 190 | |
| History of at least one symptom suggestive of COVID-19 | Yes | 16 (11.3) | 126 (88.7) | 142 | 0.000* |
| | No | 294 (42.7) | 395 (57.3) | 689 | |
| Tested positive | Yes | 5 (4.7) | 102 (95.3) | 107 | 0.000* |
| | No | 305 (42.1) | 419 (57.9) | 724 | |
| Family members tested positive | Yes | 12 (11.8) | 90 (88.2) | 102 | 0.000* |
| | No | 298 (40.9) | 431 (59.1) | 729 | |

## Discussion

Our study revealed a very high seroprevalence of COVID 19 infection in Bangalore rural district in July 2021, confirming colossal spread of SARS COV 2 in the rural areas during the second surge of COVID 19 pandemic. The findings are consistent with the preliminary reports of fourth serial Indian Council for Medical Research (ICMR) serosurvey [15]. ICMR national sero survey reported an overall national seroprevalence of 67.6% and a state (Karnataka) seroprevalence of 69.8% [16] in the survey conducted in June and July 2021. To estimate the state seroprevalence, ICMR sampled three districts of the total thirty districts and Bangalore rural district was excluded. The slight variability in the seroprevalence can be attributed to the difference in the geography, population density and social structure of sampled districts with Bangalore Rural district. Another reason for slight variation can be attributed to the fact that ICMR study included all population irrespective of vaccine status (seroprevalence among unvaccinated in ICMR study was 62%16) [17] whereas we only included the vaccine naive population.

Another important finding is the massive seroconversion of 12.4% to 62.7% of rural population in the second pandemic surge from which translates to 500000 people. This seroconversion happened mostly in two months, which is evident from the trend of new cases and total active cases in the same period [14]. Massive infection surge in such short time span in the background of limited health system capabilities had resulted in utter desperation and many preventable deaths (0.5 in March to 44.1 per 100,000 population in May 2021) in this district [14].

In the first surge, the urban poor in dense informal settlements got infected with the virus [4]. Despite high population densities, people in cities who had ventilated homes and the means to physically distance at workplace were spared in the first wave [18]. Low population density and favourable ventilation in rural landscapes spared the rural population from COVID 19 infection in the first surge [5]. However, the second surge predominantly dominated by Delta strain of SARS COV 2 [19], spread rapidly through urban and rural areas infecting the susceptible. Though rural population had the advantage of low population density and well ventilated landscapes, the virus found its way to infect most of the rural inhabitants of Bangalore rural district.

Case-to-undetected-infections ratio (CIR) is 1:8.65, that is for every one confirmed COVID 19 case, there were 8.7 cases that were not detected. This means when the district reported 59,112 [14] confirmed cases, the reality was 5,11,722 COVID 19 cases. This is of no surprise due to many reasons. Firstly, most of the people infected were asymptomatic and hence not detected. Secondly, the health seeking behaviour of people is rural areas are generally poor compared to urban counterparts. Thirdly, the limited capability of the rural health system to identify COVID 19 infections. CIR in the district was almost similar in first (1:7) and second wave (1:8.7) though the case load in second surge was almost 10 times than the first wave. This shows that the district was able to ramp up the diagnostics to meet the increased testing demand during the surge crisis [5].

Infection Fatality Rate (IFR) was calculated as 16.27 per 100,00 infections as on 13 July 2021 when compared to 12.38 per 100,00 infections as on 22 October 2020 in Bangalore rural district. The increased mortality rate can be attributed to inordinate increase in the number of cases and limited healthcare infrastructure capabilities in the rural district [5]. A poor healthcare network at the grassroot level and inadequately trusted health information channels throttled the existing health system leading to many avoidable deaths in the district during the second surge.

Though vaccine was rolled out in a prioritised manner to general public from April 2021, the pace of vaccination was no match for the velocity of viral transmission. At the end of May 2021, only 20% of the population of the district was vaccinated with at least one dose [14]. Vaccination demand was high during the pandemic surge, however the vaccine production could not meet the demand at that time. At the end of July 2021, vaccination stood at 40% for single dose and 10% for both the doses in Bangalore rural district [20]. Aggressive and strategic vaccination in the early phases could have reduced the severity of this surge, however a comparatively low active case load for 3–4 months gave a false notion of the pandemic being under control which led to vaccine complacency [18].

We could draw several implications from the findings of the study. First and foremost there is a relief that the population is heading towards herd immunity. But it is also time to introspect whether it is really a sigh of relief considering the heavy price that we paid with many lives. There cannot be a stronger voice for building a robust rural healthcare infrastructure which is able to handle the primary, secondary and tertiary healthcare needs of the community. The nature of the health infrastructure should be transformed from one that of a 'health custodian' to a that of a 'trusted partner' to improve healthcare in these communities. We have learned that 'trust' is an underrated virtue that can pay huge dividends during normalcy and in times of crisis.

Secondly, we have made a huge mistake by being vaccine complacent thinking that the pandemic is under control. Vaccinating a one billion population is a herculean challenge when we consider the inequities and diversities of our population. However, we need to devise ways to vaccinate them all, despite the challenges. The measure of our success will depend on the pace at which we deliver the vaccine to the weakest and farthest, making the society safe for lives and livelihoods.

Thirdly, we need to focus on rebuilding the damages caused by the pandemic. Mental health issues caused by COVID 19 owing to its startling pace and shocking severity needs to be addressed. The trauma of unanticipated premature deaths, the strain of locked up childhood and teens and the stress of loss of livelihoods with the background of a little or no mental health support is a painful reality in rural areas. We also need to prepare ourselves to battle the consequences of suspension of routine disease control programmes, immunisation and primary health care services during the pandemic in the coming months. It is also equally important to take care of the mental health of the healthcare providers. Even before the trauma of the surge fades, the healthcare providers are battling with neglected medical conditions of the population and combating vaccine hesitancy.

There are many strengths to this study. This is one of the earliest population-based seroprevalence study discussing the effect of first and second surge of pandemic in the rural context. Recruiting all the eligible people in the selected villages, than sampling a subset of population, calculates the true estimate with lesser potential biases. Rural areas are often less represented in publications; hence this will lend valuable learning insights and can guide local public health action.

This study has a few limitations. Firstly, we have sampled only one of the four subdistricts, so the real estimate of the district may vary slightly. However, we do not think that this can cause much variation as the population in the sub districts are homogenous and the previous serosurvey estimated similar prevalence in all the sub districts. Secondly, there is a possibility of slight underestimation of the seroprevalence since we have not included those with current infection. Thirdly, The seroprevalence was quantified using an assay that measures primarily IgG and IgM antibodies against RBD. However, The kit gives a total antibody titre without relative measures of IgG, IgM antibodies. We have analysed total antibody titre for males and

females and stratified by age. Another limitation is that the upper limit of antibody titre is reported as >250 U/ml than actual value as it was not tested with dilution.

## Conclusion

This study in a rural district of South India during the second surge showed a high seroprevalence of COVID-19 infection despite the advantages of low population density and well-ventilated rural landscapes. Case-to-undetected-infections ratio and infection fatality rate were higher than the previous serosurvey conducted in the same population during the first surge. Though the thought of achieving herd immunity comes with a relief, it would prove beneficial for us to introspect about the price–of losing many lives–we paid for it. The path ahead of us is not easy having to battle diseases that were neglected during the pandemic and recovering from the mental trauma caused by the pandemic.

Humanity's memory is notoriously short, but this is a period we must remember. The pandemic has exposed health vulnerabilities and taught us the value of health equity. Isolation and disruptions made us realise the value of social connection and solidarity. We must continue our efforts on vaccination till the weakest link is reached to drive down infections and to keep the variants at bay. We must take this opportunity to nurture people's health and build a stronger rural healthcare infrastructure. The time has come to create a more 'connected healthcare' system, where communities and hospitals interact meaningfully and seamlessly to build population health insulating against future pandemics.

## Supporting information

**S1 Table. Age- and gender-standardized seroprevalence.**
(LOG)

**S1 File. Data.**
(SAV)

## Acknowledgments

We would like to thank Mr Tata Rao for creating Epi-data for data collection, conducting preliminary analysis and his help with referencing. We also acknowledge The Kurian Foundation Trust for their support in conducting the study. We also extend our thanks to the phlebotomists Mr Jeyapal, Mr Ajith and the entire field team. We are also grateful to the village leaders, health committee members, and community members for their support and enthusiastic participation in the study.

## Author Contributions

**Conceptualization:** Carolin Elizabeth George, Leeberk Raja Inbaraj, Shon Rajukutty, Sindhulina Chandrasingh.

**Data curation:** Leeberk Raja Inbaraj, Roshni Florina Joan, Arun Karthikeyan Suseeladevi.

**Formal analysis:** Carolin Elizabeth George, Leeberk Raja Inbaraj, Arun Karthikeyan Suseeladevi.

**Investigation:** Carolin Elizabeth George, Roshni Florina Joan.

**Methodology:** Carolin Elizabeth George, Leeberk Raja Inbaraj, Shon Rajukutty.

**Project administration:** Carolin Elizabeth George, Shon Rajukutty, Sangeetha Muthuraj, Sindhulina Chandrasingh.

**Supervision:** Shon Rajukutty, Roshni Florina Joan, Sangeetha Muthuraj.

**Validation:** Carolin Elizabeth George, Leeberk Raja Inbaraj, Arun Karthikeyan Suseeladevi, Sangeetha Muthuraj, Sindhulina Chandrasingh.

**Writing – original draft:** Carolin Elizabeth George, Leeberk Raja Inbaraj.

**Writing – review & editing:** Carolin Elizabeth George, Leeberk Raja Inbaraj, Shon Rajukutty, Roshni Florina Joan, Arun Karthikeyan Suseeladevi, Sindhulina Chandrasingh.

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
