## [Editor Report · Decision Letter 0]

26 Dec 2021

PONE-D-21-30742Seroprevalence of COVID-19 infection among vaccine naïve population after the second surge (June 2020) in a rural district of South India: a community-based cross-sectional studyPLOS ONE

Dear Dr. Inbaraj,

Thank you for submitting your manuscript to PLOS ONE. After careful consideration, we feel that it has merit but does not fully meet PLOS ONE’s publication criteria as it currently stands. Therefore, we invite you to submit a revised version of the manuscript that addresses the points raised during the review process.

ACADEMIC EDITOR:

We apologize for the delay in processing this manuscript. While the study is of significant interest, there are several key issues that need to be addressed in a revised manuscript: 1. The Methods section states that seroprevalence was quantified using an assay that measures primarily IgG and some IgA + IgM antibodies against RBD. The Abstract states that IgG and IgM antibodies against RBD were quantified. This discrepancy needs to be resolved. More importantly, can the authors provide relative abundance levels of IgG and IgM antibodies? This would for obvious reasons provide key insights into the duration of infection. These analyses should either be added, or if the data is unavailable, this should explicitly be discussed as a study limitation. 2. The study was carried out in a population that hadn't received a COVID-19 vaccine. Given the timeline of the study, this could be either due to lack of access to a COVID-19 vaccine or general vaccine hesitancy. However, are other vaccination records available for these individuals? This information may help identify 2 sub-groups in the population - a) those that have completed recommended vaccination regimens for other vaccines i.e., they had likely not received a COVID-19 vaccine at the time of the study due to lack of access (potentially lower risk group), and b) those that had not completed recommended vaccination regimens even for other vaccines i.e., they likely have general vaccine hesitancy (potentially higher risk group). Are there indeed differences in seropositivity between these 2 groups? 3. Table 3 provides insights into how univariate factors are associated with seroprevalence. The insights from this table are limited as the significant univariate factors are the "obvious" ones (symptoms, positive test for individual or family members). It would be much more interesting to examine the contributions of the "non-obvious" features to seroprevalence risk in a multivariate setting (potentially also accounting for interaction terms). 4. From a presentation standpoint, the manuscript would benefit significantly from some figures/visual representations that summarize the data currently presented as tables.

We look forward to receiving your revised manuscript.

Kind regards,

Jishnu Das, Ph.D.

Academic Editor

PLOS ONE

Journal Requirements:

2. Please include additional information regarding the survey or questionnaire used in the study and ensure that you have provided sufficient details that others could replicate the analyses. For instance, if you developed a questionnaire as part of this study and it is not under a copyright more restrictive than CC-BY, please include a copy, in both the original language and English, as Supporting Information

Reviewers' comments:

Please see the comments above.

---

## [Author Response · Author response to Decision Letter 0]

6 Jan 2022

A rebuttal document responding to the comments of the academic editor is attached separately.

---

## [Editor Report · Decision Letter 1]

28 Feb 2022

Seroprevalence of COVID-19 infection among vaccine naïve population after the second surge (June 2020) in a rural district of South India: a community-based cross-sectional study

PONE-D-21-30742R1

Dear Dr. Inbaraj,

We’re pleased to inform you that your manuscript has been judged scientifically suitable for publication and will be formally accepted for publication once it meets all outstanding technical requirements.

Kind regards,

Jishnu Das, Ph.D.

Academic Editor

PLOS ONE
---

## [Editor Report · Acceptance letter]

2 Mar 2022

PONE-D-21-30742R1 

Seroprevalence of COVID-19 infection among vaccine naïve population after the second surge (June 2020) in a rural district of South India: a community-based cross-sectional study 

Dear Dr. Inbaraj:

I'm pleased to inform you that your manuscript has been deemed suitable for publication in PLOS ONE. Congratulations! Your manuscript is now with our production department. 

Kind regards, 

on behalf of

Dr. Jishnu Das 

Academic Editor

PLOS ONE